# Density-Dependent Seed Predation of *Quercus wutaishanica* by Rodents in Response to Different Seed States

**DOI:** 10.3390/ani13111732

**Published:** 2023-05-24

**Authors:** Yonghong Luo, Jiming Cheng, Xingfu Yan, Hui Yang, Yan Shen, Jingru Ge, Min Zhang, Jinfeng Zhang, Zhuwen Xu

**Affiliations:** 1School of Ecology and Environment, Inner Mongolia University, Hohhot 010021, China18234487746@163.com (Y.S.); 2School of Life Sciences, Central China Normal University, Wuhan 430079, China; 3College of Biological Science and Engineering, North Minzu University, Yinchuan 750021, China; 4Optoelectronic College, Beijing Institute of Technology, Beijing 100081, China; 5School of Ecology and Nature Conservation, Beijing Forestry University, Beijing 100083, China

**Keywords:** *Quercus wutaishanica*, seed dispersal, rodents, habitats, seed sizes, density-dependent

## Abstract

**Simple Summary:**

The predation and/or dispersal of *Quercus* seeds by rodents play an important role in the creation of the tree species. Using the tagging method, we measured the effects of density, storage method, and seed size on rodents’ predation and dispersal behavior. We found that seed survival had a significant negative density-dependent effect; that litter cover and soil burial increased seed in situ time and survival; that in situ feeding rates were significantly higher for small seeds than for large seeds; and that large seeds fed significantly further after dispersal than small seeds. These findings provide insights into the ecological characteristics of *Quercus* tree regeneration and shed light on the coexistence between rodents and different-sized seeds.

**Abstract:**

The predation and/or dispersal of *Quercus* seeds by rodents play an important role in the creation of the tree species. The present study examined the effects of community habitats on the predation and dispersal of *Quercus wutaishanica* seeds by rodents. We released seeds with densities set at 2, 4, 8, 16, and 32 seed square meter with litter cover, soil burial, and bare ground in the Liupan Mountains National Nature Reserve in the Ningxia Hui Autonomous Region, northwest China. The results showed that (1) the litter cover and soil burial significantly increased the seed survival probability compared with bare ground treatments, especially the predation in situ (PIS) (*p* < 0.05). Both the scatter hoarding (SH) and larder hoarding (LH) for litter cover and soil burial were significantly increased compared with bare ground (*p* < 0.05). (2) The large seeds are preferentially predated after dispersal and their long-distance dispersal (>5 m) was significantly greater than that of small seeds (*p* < 0.05), while small seeds are more likely to be preyed on in situ or during short-distance dispersal (<3 m). (3) The *Q. wutaishanica* seed predation by rodents increased at a high density rather than at a low density, indicating a negative density-dependent predation. These findings provide insights into the ecological characteristics of *Quercus* tree regeneration and shed light on the coexistence between rodents and different-sized seeds.

## 1. Introduction

Rodent-mediated seed predation plays an important role in seedling establishment, population dynamics, and species coexistence [1,2]. When the seeds mature and scatter, small rodents hoard higher quantities of seeds in case of food shortages [3]. The behavior of rodents in hoarding food usually takes two forms: (1) larder hoarding, which refers to animals that store all their food in one or a few caches or nests; (2) scatter hoarding, which means that rodents store food in many caches with a small amount of food in each cache [4]. The former could lead to catastrophic consequences once food is stolen, and, on the contrary, the latter may be more beneficial for the natural regeneration of plants [5,6,7]. Even if some of the hoarded food is eaten by predators or thieves, some seeds may be forgotten or scattered, allowing for successful seedling establishment [8,9].

The Janzen–Connell theory predicts that seed density may change the predators’ selection preferences [10,11]. High-density seeds may cause large amounts of predation by animals or infection by specific pathogens, thereby reducing the chance of seedling establishment [12,13]. Conversely, the farther the seed is from the parent tree or the lower the seed density, the greater the chance of escape and the more conducive it is to seedling building [13]. Moreover, it is likely that the direction, as well as the intensity, of selection preferences including predation, dispersal, and hoarding will vary with the seed size [14,15,16,17]. Generally, large seeds may disperse farther and are likely to be buried by predators, which is conducive to seedling establishment, while small seeds may be eaten by animals due to their lower nutritional benefits, which are not worth the longer investment, and the distance to be transported is shorter [2,18].

Aside from the seeds’ size and density, both the predators’ selection preferences and the fate of seeds are highly susceptible to heterogeneous habitats [3]. There is evidence that the seeds of some trees are easier to find on bare ground than buried [19,20]; thus, the litter cover and soil burial could reduce the risk of seed predation and thereby increase the chance of seedling establishment [20]. Moreover, the positive effect of burial is also reflected: (1) to mitigate the impact of adverse environmental factors (e.g., high temperature and low moisture), or (2) shape a mechanical barrier by changing the chemical environment of the seeds, which contributes to seed germination and seedling establishment [20].

*Quercus wutaishanica* is the dominant species of the warm, temperate, deciduous, and broad-leaved forest in northwest China [3]. The *Q. wutaishanica* seed varies in seed size at both interspecific and intraspecific levels, which in turn, affects the hoarding and predation preferences of rodents [20]. Previous studies of *Q. wutaishanica* have shown that seed predation, dispersal, and hoarding by small mammals prior to germination are dependent on seed size [2,20], but density-dependent seed predation has not been well studied, especially when the seeds are in different states. Here, we describe a field experiment focusing on the *Q. wutaishanica* seeds in three states to explore the influence of seed sizes and their density on rodents’ hoarding and predation behaviors. We expected that (1) rodent-mediated seed dispersal on bare ground may be faster than litter cover and soil burial. (2) Seed size effects on hoarding and predation preferences of rodents. (3) Rodent’ hoarding and predation behaviors dependent on seed density.

## 2. Material and Methods

### 2.1. Study Sites

This study was conducted in the Liupanshan National Natural Reserve (35°15′ N to 35°41′ and 106°09′ to 106°30′ E) in the Ningxia Hui Autonomous Region, northwest China. The area at the edge of the northern agriculture–pastoral ecotone and the semi-humid to semi-arid region in the warm temperate zone. It is hot and rainy in the summer, while dry and cold in the winter. The total annual precipitation is about 767 mm with 60% precipitation from June to September, and the annual evaporation is about 1426 mm [3,20]. The annual mean temperature is 5.8 °C, with the extreme high and low temperatures being about 30 °C (July) and about −26 °C (January) [3,20]. The soil type is mainly grey cinnamon. The main woody vegetation includes trees such as *Q. wutaishanica*, *Tilia paucicostata*, *Pinus tabulaeformis*, *Betula platyphylla*, etc., and shrubs including *Rubus* spp., *Crataegus kansuensis*, *Amygdalus davidiana*, *Cotoneaster multiflorus*, etc. The herb mainly comprised *Brachypodium sylvaticum*, *Carex tristachya*, *Phlomis umbrosa*, etc. Shrublands and secondary forests of *Q. wutaishanica* are the main types of vegetation at the study site [3,20]. *Q. wutaishanica* seeds matured from August to September. Natural regeneration of this species in this area was often influenced by rodents, and several rodent species, including *Niviventer confucianus*, *Mus musculus*, *Sciurotamias davidianus*, *Apodemus agrarius*, and *Microtus fortis* [3,20].

### 2.2. Seed Collection

*Q. wutaishanica* seeds were collected from Mt. Liupan on 18 September 2017. In order to remove the damaged seeds (worm-eaten and moldy), they were stored in the laboratory for a week, and then seeds were separated based on their sizes (the weight of large and small seeds is (mean ± standard deviation (SD), 3.05 ± 0.38 g) (n = 100) and (1.46 ± 0.27 g) (n = 100), respectively.

### 2.3. Marking and Labeling of Seeds

A 7.5 cm long, 0.8 mm diameter copper wire was fixed at a pink plastic label with size of 2 cm × 1.2 cm (containing the mass of the copper wire (0.17 ± 0.002 n = 100) g). Each label was written in pencil with relevant information such as plot location, seed size, seed density, and habitat (bare ground, litter cover, and soil burial), so as to search for and record the fate of the seed [21]. We used an electric drill with a drill bit of 1 mm to make a small hole at the end of the seed away from the germ, and then secured the seed with the prepared label [21,22].

### 2.4. Experimental Design

On 8 April 2018, in the secondary forest of *Q. wutaishanica* in Mt. Liupan Nature Reserve, five transects (90 m × 20 m: length × width) were selected. Each transect was considered as density treatments with ratios set as 2 seed Sq. m, 4, 8, 16, and 32 Sq. m, respectively. We set up three plots (2 m × 2 m, each plot was equally spaced ~25 m) as three bare ground treatments from the bottom to the top of the hillside in the midline area of each transect. Furthermore, three litter cover treatments with a size of about 2 m × 2 m are set at five meters from the bare ground as three replicates, and the thickness is about 2 cm. In each experimental transect, we selected three plots buried with 2 cm soil, which had similar sizes to the bare ground and soil burial treatments (each 5 m away from litter cover treatments) (Figure 1). The large and small seeds (1:1) were evenly released at each plot. Total seeds: 5 densities (2, 4, 8, 16, and 32) × plot area (4 Sq. m) × 3 states (bare ground, litter cover, and soil burial) × 3 repetitions = 2232 seeds.

### 2.5. Field Investigation

We recorded the number of seeds that were preyed on, dispersed, and buried on the 1st, 2nd, 3rd, 5th, 15th, 30th, and 60th days of release plots. We searched and recorded the seed label in transects, and measured the distance from the seed to the falling point using a tape. Here, we recorded the number of seeds retained, the number of in situ preying, the number of preying after dispersal, and the number of stored seeds after dispersal. Dividing the fate of seeds into six types: retained in situ (RIS), predation in situ (PIS), scatter hoarding (SH), larder hoarding (LH), predation after dispersal (PAD), and lose (L).

### 2.6. Data Analysis

We used Cox regression to analyze the seed survival probability for habitats, densities, and seed sizes during different observation periods. Cox regression was fitted using “Surv” function in the “survival” package. We constructed generalized linear mixed-effects models (GLMMs) for seed retention rate remaining in the experimental transect after the end of the observation period, with fixed factors for the seed sizes, densities, and states and random effect terms for plots and observation period. The random-effects term of plot and observation period were included to allow us to measure variations of plot-to-plot and different observation period, which are not explained by fixed factors. GLMMs were fitted using “glmer” function (family = “beta”) in the “lmer4 package” [23]. To test the impact of the interaction between seed size, density, and state on seed retention rate, we used the “anovo function”. These analyses were performed with R 4.2.0 (R Development Core Team; http://r-project.org; accessed on 2 November 2022).

One-way ANOVA with SPSS (Version 21.0) was used to evaluate the effect of the seed states, sizes, and densities on seed fates, including RIS, PIS, SH, LH, PAD, and L. This method was also used to determine the influence of seed states, sizes, and densities on the dispersal distance, and the least significant difference method (LSD) was used to detect differences. The normality test and standardization of data are conducted in SPSS 21.0. We used SigmaPlot (Version 12.5) to make all figures.

## 3. Results

### 3.1. Quercus Wutaishanica Seed Dynamics

Cox regression analysis showed that rodents prey on seeds the fastest on bare land, and the predation of rodents was significantly inhibited after soil covering (Wald = 12.56, *p* = 0.045) (Figure 2a). During the observation period, the survival probability of small seeds was significantly higher than that of large seeds (Wald = 7.78, *p* = 0.032) (Figure 2b). The survival probability was significantly different among different densities (Wald = 24.59, *p* = 0.005) (Figure 2c), and from high to low, it was 2, 4, 8, 16, and 32 Sq. m, respectively.

### 3.2. Quercus Wutaishanica Seed Fates

There is no significant difference between litter cover and soil burial for the RIS (Figure 3a). Significant differences were observed for the PIS and SH in different treatments (*p* < 0.05) (Figure 3a). Soil burial and litter cover treatments were significantly higher than bare ground for the LH (*p* < 0.05). Soil burial and litter cover treatments were significantly higher than bare ground for the PAD (*p* < 0.05). There is no significant difference in the lost seeds among different treatments (Figure 3a).

In terms of the RIS, there were no significant differences between the large and small seeds. The PIS for large seeds was significantly lower than that for small seeds (*p* < 0.05) (Figure 3b). In contrast, the SH for large seeds was significantly higher than that for small seeds (Figure 3b) (*p* < 0.05). There were no significant differences in the rates of LH, PAD, and L for large and small seeds (Figure 3b).

The RIS and PIS of seeds decreases with increasing seed density (Figure 3c). In contrast, the SH, LH, and PAD of seeds increase with increasing seed density (Figure 3c). Seed loss rates were significantly lower at 2 seeds Sq. m and 32 seeds Sq. m than at the other three densities (*p* < 0.05) (Figure 3c).

### 3.3. Dispersal Distance

Seeds were carried near the release point (<1 m), and the soil burial treatment was significantly higher than those placed in bare ground and litter cover (*p* < 0.05) (Figure 4a). Seeds were carried 1–3 m to the release point, and the litter cover was significantly higher than the bare ground and soil burial treatments (*p* < 0.05) (Figure 4a). Seeds were carried 3–5 m to the release point, and there was no significant difference between the three placement methods (Figure 4a). Seeds were carried 5–10 m and >10 m to the release point, and the bare ground treatment were significantly higher than those from the soil burial and litter cover treatments (*p* < 0.05) (Figure 4a).

Near the seed release point (<1 m), we observed that the retention rate of small seeds was significantly higher than that of large seeds (*p* < 0.05) (Figure 4b). Within 1–5 m of the seed release point, there was no significant difference in the seed retention rate between different-sized seeds (Figure 4). When the dispersal distance is 5 m away from the release point, the retention rate of large seeds is significantly higher than that of small seeds (*p* < 0.05) (Figure 4b).

The percentage of seeds carried relatively close to the release site (>1 m, 1–3 m, and 3–5 m) increased with the increasing placement density (Figure 4c). Conversely, the percentage of seeds carried relatively far from the release site (5–10 m and >10 m) decreased with increasing placement density (Figure 4c).

The fixed effects, including seed sizes, densities, and states, as well as their interactions, explained 32.25% of the seed retention rate, and the densities and states significantly influenced the seed retention rate (*p* < 0.05). The random effects, including plots and observation time, explained 22.58% and 15.89% of seed retention rate, respectively (Table 1).

## 4. Discussion

### 4.1. Microhabitat Affects the Predation and Dispersal Behavior of Rodents

Oak seeds rely mainly on internal rodents for dispersal and then for seed germination, seedling establishment, and seedling survival to ensure successful regeneration [24,25]. However, the heterogeneous habitat may affect the distribution of rodents and then affect their foraging activities, ultimately affecting the dispersal and survival of seeds [26]. In this study, we found that litter cover and soil burial significantly increased the seed retention ratio compared with bare ground treatments (Wald = 12.56, *p* = 0.05) (Figure 2a, Table 1), especially the PIS (*p* < 0.05) (Figure 3a). This may be the result of the trade-off between predation risk and profit [26]. Given that the cost of capturing prey is high, predators should increase their foraging success by focusing on prey species that are easier to find [27]. However, covering litter and burying soil will increase the time rodents need to search for seeds making them more vulnerable to predation [3]. Conversely, rodents could detect food quickly when the obstacle affecting rodents’ foraging activities is removed, thus reducing the cost of hunting prey to a certain extent [3].

It is noteworthy that no matter what seed states, rodents prefer to consume seeds in situ, and the seed percentage of PIS was higher than the RIP, SH, LH, PAD, and lose. The foraging decision of rodents is the result of the comprehensive weighing of many factors [28]. The rodent predation risk is increased if the seed handling time is prolonged [29]. Therefore, rodents choose to prey on the seeds in situ in order to improve the net income per unit time [30]. Furthermore, we found that the retention percentages of litter cover and soil burial treatments were significantly higher than bare ground for the PAD (*p* < 0.05). This may be another trade-off between the time of searching for seeds and the risk of being preyed by others; in other words, if rodents need to invest a lot of energy in their search for seeds, there is no doubt that predation after dispersal is a safe behavior strategy [30]. Meanwhile, the positive effects of litter and soil are reflected in the hoarding behavior of rodents. For example, compared with bare ground, the SH and LH were significantly increased (*p* < 0.05) (Figure 3a). Rodents could preferentially store or hoard seeds when the cost is greater than the hoarding, because the former means investing more time and energy, losing the opportunity to prey on other foods, and increasing the risk of being predated by natural enemies [2,31]. In the evolutionary sense, the benefit from rodents hoarding seeds is far greater than the cost of the seeds that they prey on. Because once the seeds escape from predation or are forgotten, they may germinate successfully; even some discarded or partially consumed seeds after being hoarded may potentially establish seedlings [22,32].

### 4.2. The Preference of Rodents for Seeds of Different Sizes

The optimal foraging theory predicts that predators prioritize catching large seeds with sufficient nutrients [33]. The detection of *Q. wutaishanica* seed selection by rodents in our study was consistent with the theoretical expectation (Figure 2b). This supports the previous findings of Zhang et al. and Celis-Diez et al. [2,20]. Rodents can estimate the seeds’ qualities, in order to maximize the energetic intake and/or to minimize the searching time. Seed predators preferentially consume large seeds that contain greater nutrient [24,25]. By extension, rodents prefer to prey on large seeds after dispersal, while small seeds are more likely to be preyed on in situ (Figure 3b), ultimately resulting in different sizes of seeds having the same dispersal fitness [2].

In addition, dispersal distance can also reflect rodents’ preferences for different seed sizes [34]. We found that the retention rate of small seeds was significantly higher than that of large seeds near the seed release point (<1 m) (*p* < 0.05), and most of the dispersal seeds were concentrated within 5 m, while the retention rate of large seeds was significantly higher than that of small seeds when the dispersal distance was 5 m away from the release point (*p* < 0.05) (Figure 4b). This result is similar with that reported by Xiao et al., who found that most of the seeds of *Q. variabilis* are transported within 6 m by rodents [21]. Our finding also conforms to the prediction results of the rapid sequestering hypothesis [7]; that is, the seed with lower nutrients is stored near the resource pool, while the seed contained higher energy is stored far away from the resource pool [4].

### 4.3. Negatively Density-Dependent

In general, rodents search for food by their acute sense of smell [12]. Stronger olfactory signals from high-density food seed release sites are conducive to successful search by rodents, and once predators find one food cache site, they may be more likely to search for the next one in the adjacent area. [1,35]. We found that the rodents’ foraging is negatively density-dependent (especially, the preference for the *Q. wutaishanica* seed is higher at a high density than at a low density, while the RIS and PIS of seeds decrease with increasing seed densities. In contrast, the SH, LH, and PAD of seeds increase with increasing seed densities (Figure 2c and Figure 3c and Table 1). Negatively density-dependent foraging was also found in community levels, and these similar studies speculated that this predation was most likely to occur when a certain prey type was more abundant [36,37]. For example, when glabrous plants were abundant, hairy plants incurred less herbivory by *Phaedon brassicae* [38], the storage of *Juglans regia* and *Castanea mollissima* with similar apparent perceived value by scattered storage animals was affected by their abundance [39]. The negatively density-dependent foraging of rodents may be driven by the number of prey individuals [40]. In other words, high-density seeds emit higher concentrations of seed odor and provide more olfactory cues to rodents, whereas low-density seeds would be forgotten or ignored by the rodents. [41,42]. Of course, the satiety effect should not be ignored either, especially at high densities. At that time, seeds are consumed more by rodents, but those that are not attacked due to satiation can survive [43]. Therefore, seed predation mediated by rodent shows a negatively density-dependent result in our study, which could alleviate the loss of rare seeds and promote seeds to escape. As a result, they are more likely to successfully germinate and establish seedlings.

In this study, we found that, after taking into consideration the differences among plots and observation time, the random effects could explain 32.25% of the seed retention rate (Table 1), which implied that other factors besides predation mainly drove seed dynamics, even though rodent-mediated seed predation seems very important in determining the short-term seed fates. This is because (1) our study focused on the overall effects across all plots on seed predation and dispersal. However, the microhabitat surrounding the seed included things such as slope position and slope degree, which may affect the predation behavior of rodents [3]. (2) Some rodents prefer to prey on one particular type of prey while others prefer to prey on alternative prey, or size selection may even be related to the size of the rodents [20]. (3) We have not ruled out the influence of secondary dispersal, especially for seeds of different characteristics. Because there is considerable variation in the seed structure, both among and within species as well as within individuals [2]. Although some studies concerning how this variation affects the primary dispersal of seeds have been conducted, little is known about the impact of this variation on secondary dispersal, by any vector; hence, understanding the relationship between the seed structures and secondary dispersal could reveal the adaptations shown by individual plants [1].

## 5. Conclusions

The results of our study suggest that litter cover and soil burial could prevent rodents from preying on seeds, while this protective effect only delayed the time for rodents to search for seeds to some extent. Once rodents find the release site, the seeds will be quickly preyed on or transported. Of course, scatter hoarding, forgotten after dispersal, or partially consumed seeds, may germinate and establish seedlings. Moreover, rodents prefer to prey on large seeds after dispersal, while small seeds are more likely to be preyed on in situ, which has implications for the potential evolution of different-sized seeds. More importantly, we found that rodent foraging is negatively density-dependent; that is, the preference for the *Q. wutaishanica* seed is higher at a high density than at a low density. However, rodent-mediated seed predation is a complex ecological process that includes a series of successive stages, and in particular, the secondary dispersal driven by rodents may be more complex due to the complexity of different study areas and the diversity of predators; therefore, future studies should especially focus on the relationship between secondary dispersal and density dependence.

## Figures and Tables

**Figure 1 animals-13-01732-f001:**
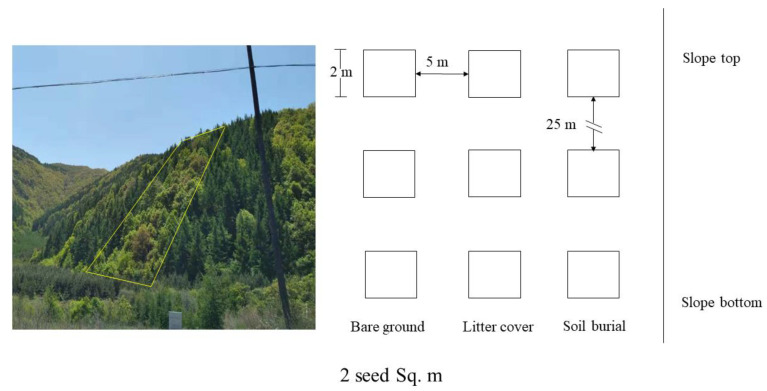
Distribution diagram of release plots in sample transect (take density of 2 seed square meter (Sq. m) as an example).

**Figure 2 animals-13-01732-f002:**
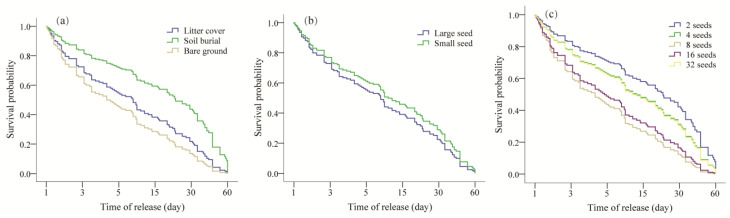
Seed removal dynamics for *Quercus wutaishanica* at seed release sites. (**a**) Habitats; (**b**) seed sizes; and (**c**) densities.

**Figure 3 animals-13-01732-f003:**
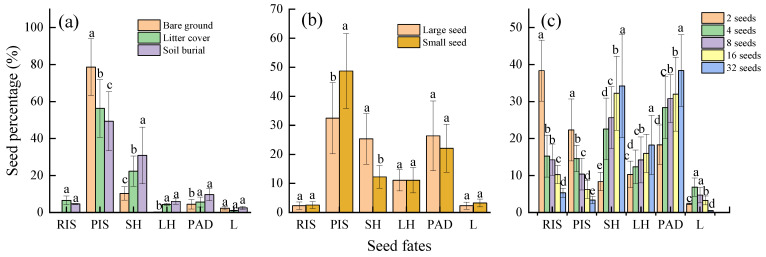
The fate of *Quercus wutaishanica* seed. RIS, retained in situ; PIS, predation in situ; SH, scatter hoarding; LH, larder hoarding; PAD, predation after dispersal; and L, lose. Note: (**a**–**c**) respectively indicates the effect of storage method, seed size and density on seed fates, with different lowercase letters on the bars indicating differences between percentages of seeds under the same fate.

**Figure 4 animals-13-01732-f004:**
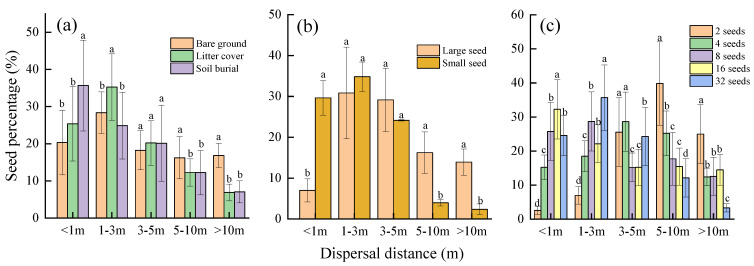
The fate of *Quercus wutaishanica* seeds transplanted in the experiment. Note: (**a**–**c**) indicates the effect of storage method, seed size and density on seed dispersal distance, respectively, and different lowercase letters on the bars indicate significant differences in the same dispersal distance.

**Table 1 animals-13-01732-t001:** Parameter estimates and variance predictions in mixed models with a random effect used to test for effects of fixed factors and random factors on seed retention rate.

	Covariates	Standardized Coefficients	*p*	*R* ^2^
Fixed effects	Seed sizes	−0.433	0.048	0.3225
	Densities	−0.224	0.035	
	States	−0.025	0.001	
	Seed sizes × Densities	−0.459	0.053	
	Densities × States	−0.763	0.121	
	Seed sizes × Densities × States	0.532	0.068	
Random effects		Variance components	*R* ^2^	
	Plots	0.0058	0.2258	
	Observation time	0.0037	0.1589	

*R*^2^, the explanatory power of the covariates and random factor, not the traditional *R*^2^.

## Data Availability

Not applicable.

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
