# Peer review of "Density-Dependent Seed Predation of *Quercus wutaishanica* by Rodents in Response to Different Seed States"

_animals, 2023, doi:10.3390/ani13111732_

Round 1

Reviewer 1 Report (Previous Reviewer 3)

We are satisfied with the new changes presented by the authors in the manuscript

Author Response

Dear Editors and Reviewers:

Thank you for your letter and for the reviewers’ comments concerning our manuscript entitled “Density-Dependent Seed Predation of Quercus wutaishanica, by Rodents in Response to Different States”(ID: animals-2385798). Those comments are all valuable and very helpful for revising and improving our paper, as well as the important guiding significance to our researches. We have studied comments carefully and have made correction which we hope meet with approval. Revised portion are marked in red in the paper. The main corrections in the paper and the responds to the reviewer’s comments are as flowing:

Responds to the reviewer’s comments:

Reviewer #2:

Abstract:

1 Comment: Sq. m: Write the full name the first time it appears.

Change the last sentence as follow:

These findings provide insights into the ecological characteristics of Quercus tree regeneration and shed light on the coexistence between rodents and different-sized seeds.

Response: We have replaced Sq. m with the full name.

We have changed the last sentence to the sentence suggested by the reviewer “These findings provide insights into the ecological characteristics of Quercus tree regeneration and shed light on the coexistence between rodents and different-sized seeds.”

Introduction

  1. In 34, "animals" could be replaced with "rodents" for more specificity.

Response:We have replaced animals with rodents.

  1. In 37-38, the phrasing is somewhat awkward. A possible revision could be:"Even if some of the hoarded food is eaten by predators or thieves, some seeds may be forgotten or scattered, allowing for successful seedling establishment."

Response: We fully accept the reviewer's suggestion and have replaced in 37-38 with"Even if some of the hoarded food is eaten by predators or thieves, some seeds may be forgotten or scattered, allowing for successful seedling establishment."

Material and methods

  1. Line 126 Dividing the fate of seeds into five types; Line 142 evaluate the effect of the 142 seed states, sizes, and densities on seed fates, include RIS, PIS, SH, LH, PAD, and L” it is six types……. . Two mismatch descriptions, please revise them.

The font size of the content in the experimental design is larger than the rest.

Response: Thank you for your reminder, we have amended.

  1. Line 138. The glmer function is available in the lme4 package in R. https://www.rdocumentation.org/packages/lme4/versions/1.1-32/topics/glmer

The lmerTest package is an R package that provides p-values for linear mixed-effects models fit using the lmer function in the lme4 package. It also offers other statistical tests and diagnostic tools for linear mixed effects models. The lmerTest package is particularly useful when dealing with complex and nested data structures, such as longitudinal or clustered data.

Please revise the Data analysis

Response: Thanks to the reviewers for their patience in explaining the difference between the lme4 package and the lmerTest package, we have reworked the data analysis according to the reviewers' suggestions.

Results

6 Figure 2 is vague, please redo it.

Figure 3 Seed fates.

Response: We have redone Figure 2 and modified Figure 3.

Discussion

Clarify the language: Some of the sentences are a bit complex and difficult to understand. Consider simplifying the wording to make it easier to follow.

Use active voice: The paragraph contains a lot of passive voice sentences. Try to rewrite them in an active voice to make the writing more engaging and direct.

Break up long sentences: Some of the sentences are quite long and could be broken up into smaller, more manageable chunks. This will make the writing easier to read and understand.

Response:We have revised the discussion section in line with the reviewers' suggestions.

We have also written simple summary, as suggested by the assistant editor, and have done our best to reduce repetition in the first manuscript.

We appreciate for Editors/Reviewers’ warm work earnestly, and hope that the correction will meet with approval.

Once again, thank you very much for your comments and suggestions.

Reviewer 2 Report (Previous Reviewer 1)

Thanks for authors' efforts to revise the manuscript. They have addressed almost my concerns. I only mentioned several general minor issues that might improve the quality of the MS:

Abstract

Sq. m: Write the full name the first time it appears.

Change the last sentence as follow:

These findings provide insights into the ecological characteristics of Quercus tree regeneration and shed light on the coexistence between rodents and different-sized seeds.

Introduction

In 34, "animals" could be replaced with "rodents" for more specificity.

In 37-38, the phrasing is somewhat awkward. A possible revision could be: "Even if some of the hoarded food is eaten by predators or thieves, some seeds may be forgotten or scattered, allowing for successful seedling establishment."

Material and methods

“Line 126 Dividing the fate of seeds into five types; Line 142 evaluate the effect of the 142 seed states, sizes, and densities on seed fates, include RIS, PIS, SH, LH, PAD, and L”  it is six types……. . Two mismatch descriptions, please revise them.

The font size of the content in the experimental design is larger than the rest.

Line 138. The glmer function is available in the lme4 package in R. https://www.rdocumentation.org/packages/lme4/versions/1.1-32/topics/glmer

The lmerTest package is an R package that provides p-values for linear mixed-effects models fit using the lmer function in the lme4 package. It also offers other statistical tests and diagnostic tools for linear mixed effects models. The lmerTest package is particularly useful when dealing with complex and nested data structures, such as longitudinal or clustered data.

Please revise the Data analysis

Results

Figure 2 is vague, please redo it.

Figure 3 Seed fates.

Discussion

Clarify the language: Some of the sentences are a bit complex and difficult to understand. Consider simplifying the wording to make it easier to follow.

Use active voice: The paragraph contains a lot of passive voice sentences. Try to rewrite them in an active voice to make the writing more engaging and direct.

Break up long sentences: Some of the sentences are quite long and could be broken up into smaller, more manageable chunks. This will make the writing easier to read and understand.

Author Response

Dear Editors and Reviewers:

Thank you for your letter and for the reviewers’ comments concerning our manuscript entitled “Density-Dependent Seed Predation of Quercus wutaishanica, by Rodents in Response to Different States”(ID: animals-2385798). Those comments are all valuable and very helpful for revising and improving our paper, as well as the important guiding significance to our researches. We have studied comments carefully and have made correction which we hope meet with approval. Revised portion are marked in red in the paper. The main corrections in the paper and the responds to the reviewer’s comments are as flowing:

Responds to the reviewer’s comments:

Reviewer #2:

Abstract:

1 Comment: Sq. m: Write the full name the first time it appears.

Change the last sentence as follow:

These findings provide insights into the ecological characteristics of Quercus tree regeneration and shed light on the coexistence between rodents and different-sized seeds.

Response: We have replaced Sq. m with the full name.

We have changed the last sentence to the sentence suggested by the reviewer “These findings provide insights into the ecological characteristics of Quercus tree regeneration and shed light on the coexistence between rodents and different-sized seeds.”

Introduction

  1. In 34, "animals" could be replaced with "rodents" for more specificity.

Response:We have replaced animals with rodents.

  1. In 37-38, the phrasing is somewhat awkward. A possible revision could be:"Even if some of the hoarded food is eaten by predators or thieves, some seeds may be forgotten or scattered, allowing for successful seedling establishment."

Response: We fully accept the reviewer's suggestion and have replaced in 37-38 with"Even if some of the hoarded food is eaten by predators or thieves, some seeds may be forgotten or scattered, allowing for successful seedling establishment."

Material and methods

  1. Line 126 Dividing the fate of seeds into five types; Line 142 evaluate the effect of the 142 seed states, sizes, and densities on seed fates, include RIS, PIS, SH, LH, PAD, and L” it is six types……. . Two mismatch descriptions, please revise them.

The font size of the content in the experimental design is larger than the rest.

Response: Thank you for your reminder, we have amended.

  1. Line 138. The glmer function is available in the lme4 package in R. https://www.rdocumentation.org/packages/lme4/versions/1.1-32/topics/glmer

The lmerTest package is an R package that provides p-values for linear mixed-effects models fit using the lmer function in the lme4 package. It also offers other statistical tests and diagnostic tools for linear mixed effects models. The lmerTest package is particularly useful when dealing with complex and nested data structures, such as longitudinal or clustered data.

Please revise the Data analysis

Response: Thanks to the reviewers for their patience in explaining the difference between the lme4 package and the lmerTest package, we have reworked the data analysis according to the reviewers' suggestions.

Results

6 Figure 2 is vague, please redo it.

Figure 3 Seed fates.

Response: We have redone Figure 2 and modified Figure 3.

Discussion

Clarify the language: Some of the sentences are a bit complex and difficult to understand. Consider simplifying the wording to make it easier to follow.

Use active voice: The paragraph contains a lot of passive voice sentences. Try to rewrite them in an active voice to make the writing more engaging and direct.

Break up long sentences: Some of the sentences are quite long and could be broken up into smaller, more manageable chunks. This will make the writing easier to read and understand.

Response:We have revised the discussion section in line with the reviewers' suggestions.

We have also written simple summary, as suggested by the assistant editor, and have done our best to reduce repetition in the first manuscript.

We appreciate for Editors/Reviewers’ warm work earnestly, and hope that the correction will meet with approval.

Once again, thank you very much for your comments and suggestions.

This manuscript is a resubmission of an earlier submission. The following is a list of the peer review reports and author responses from that submission.

Round 1

Reviewer 1 Report

· This study evaluates the rodent-mediated seed predation of Quercus wutaishanica in response to different habitats, seed sizes, and densities. The results suggest that rodents prefer to scatter hoarding and larder hoarding for litter cover and soil burial compared with the bare ground, which significantly increased seed survival probability. The large seeds are preferentially predated and remove a long-distance than that of small seeds. The Q. wutaishanica seed predation by rodents increased at high density, indicating negative density-dependent predation occurred. In general, the field results are of interest, but there are some puzzling and flawed problems in this article; for example, the conclusions seem unreliable because the data analysis is unreliable, and many data that should be analyzed are ignored by the authors..

· 

.

Introduction section

The author mentioned the density dependence and different habitats in the title of MS. But in the INTRODUCTION section even throughout the text, the author focuses much on seed size. Especially in the results of the paper, seed density is only mentioned in the survival analysis. I suggest the authors reconsider what is the key point in this study, seed density or seed size. If it is seed density, I don’t think the authors make it clear why study the density-dependent seed predation of Quercus wutaishanica by rodents in different habitats in the introduction.

I suggested as follow: In the first paragraph, the authors describe the rodent-mediated seed predation mechanism and its importance for plant recruitment. Then the Janzen-Connell hypothesis should be mentioned immediately in the second paragraph, with a list of relevant research advances and limitations on the density-dependent predation behavior in rodents and tree seeds. In the third paragraph it is important to mention the progress of research on the effects of different habitats on seed fates by rodents and then combine the habitat and seed density. The next paragraph introduces the research subject, methods, issues, and related hypotheses.

Method section

How to reject the damaged seeds (infested and moldy)?

It is not original for the seed marking and labeling method, and references need to be added.

The Experimental design is very messy and confusing, so that I can not understand how to carry out the experiment, such as I really can not understand how to set subplots in 90m*20m transect? how many seeds were released? What is the depth of litter cover and soil burial for seeds?

The author set three treatments, and called them habitats. I don't think it can be defined as habitats. The treatments are just the three states of seeds, cannot be called habitats, because under natural conditions, seeds only exist on the surface and can not enter the soil by themself. Therefore, buried seeds removed by rodents is not same as surface seeds, it will be more suitable called pilferage.

Why did you set each interval 5m horizontally and 25m vertically? The results show that a proportion of the seeds were dispersed in distances greater than 5m. It interferes with the independence of each repetition in such a distance spacing between seed stations.

What is the slope and orientation of the experimental site? Did the authors survey the population size of the rodents? The relevant literature suggests that slope affects rodent population size(Lee et al., 2019). How can these disturbing factors be removed?

Reference:

Lee, J.-K., Hwang, H.-S., Eum, T.-K., Bae, H.-K., & Rhim, S.-J. (2019). Cascade effects of slope gradient on ground vegetation and small-rodent populations in a forest ecosystem. Animal Biology, 70, 1–11. https://doi.org/10.1163/15707563-20191192

Statistical analysis

Did you consider the interaction of factors in the survival analysis? How do you define seed survival in GLMMs? Is the data normal distribution, or is it Binomial, Poisson, or Negative Binomial? Seed size and habitat are crossover designs, are there interactions between seed size and habitat? 

The “lmerTest” package provides p-values in type I, II, or III ANOVA and summary tables for linear mixed models. Does it is appropriate for GLMM analysis in the “lmertest” package? Is it correct that the GLMMs were fitted using the “glmer” function in the “lmerTest” package, not the “lmer”or “nlme” or”glmmTMB” package? Is there an available for “family = beta” in the r package 'lmerTest'?

The statistical inference of these linear mixed models is based on Type I, II, or III ANOVA. Is the degree of freedom properly estimated (e.g. Satterthwaite's degrees of freedom method, available in r package 'lmerTest')?

Reference

https://cran.microsoft.com/snapshot/2018-11-14/web/packages/lmerTest/lmerTest.pdf

https://www.rdocumentation.org/packages/lme4/versions/1.1-31/topics/lme4-package. https://www.rdocumentation.org/packages/stats/versions/3.6.2/topics/family.

Results

Data analysis is not in accordance with the principles of statistics, so the authors' conclusions may be unreliable. As (F168,301=28.869, P<0.001 and F66,82=12.982, P<0.001)....etc., what is it the number with the subscript 168,301? The subscript number mean degree of freedom, but what you mean?

It is obvious that there may be interactions between the factors, but the author ignores them in the analysis.

In the presentation of seed fate and seed dispersal distance, why did you not show the seed fates at different seed densities or sizes?

Discussion

The discussion section mentions that seed odor appeals the seed predation by rodents, but in this experiment, the author set up soil cover, and foliage cover, which inhibited seed odor to some extent, so when the seeds are dense but buried by the soil? What is the fate of seeds? Therefore, according to the results you have shown, how can it be determined the change in seed survival results from a change in habitat or seed density with changing odor concentrations? There are many factors in this paper, so it is not clear how the role of seed density affects seed predation as habitat changes.

In addition, the authors do not address the results and discussion of the dependence of rodent hoarding behavior on seed density, mentioned in the last paragraph of the introduction.

L239: The text does not show the proportion of seed predation by rodents at different densities. How did the authors conclude that seed predation mediated by rodents negatively shows density-dependent?

Conclusion

“Of course, forgotten seeds after dispersal may germinate and establish seedlings.” Have you monitored the seedlings?

“Have an important impact on the regeneration and distribution of plant populations”. “Which is beneficial to the tree regeneration of Q. wutaishanica Repetition of conclusions.

Therefore, I suggest the authors reconsider the main message and rewrite the conclusion.

Specific comments.

Line 17: scatter hoarding (LH) and larder hoarding (LD); Line 113: scatter hoarding (SH), larder hoarding (LH). Please standardize abbreviations

Line 110: wo or we?

Line 167: Figure 4 shows the distance of the seeds, but the caption shows “the fate of the seeds……”

Line 267: What is the net effect? You did not show in results.

Reviewer 2 Report

I found the presented manuscript interesting and scientifically sounding, and the research was well designed and conducted. Below are detailed notes to improve the quality of presentation.

 line 60 onwards: the authors formulated expectations that act as a research hypothesis. Although this is a common form, the correct hypothesis should be formulated in a way that will lead to its falsification, not confirmation. Falsification is a strong form of testing a hypothesis. Authors may or may not take this my suggestion into account.

line 79: If one of the goals of the work is to detail the interactions between rodents and seed dispersion, and thus the influence of rodents on natural regeneration, a more neutral term is advisable.

Line 93. how did you ensure that such a significant addition to the original seeds would not affect rodent behaviour, at least through species selection of rodents (differential response to the presence of the label)? This requires an explanation.

line 159: “Dispersal distance”. I think a more detailed analysis of fate of small seeds vs. large seeds would be welcome. Retention rate (which I understand as the number of seeds left uneaten, but also not hoarded?) is one thing, but I also expect a clear analysis on the level of eating/hoarding of small and large grains. This is important because it is a natural seed selection made by rodents, that could lead to some population conclusions. It seems to me that the conducted analysis suggests some of the conclusions in the discussion, but not all of them, and it would be useful to supplement them here.

Line 256: please add at least a little bit of information about what species of rodents in this particular experiment may have been feeding on the seeds.

Line 256: I also suggest an attempt to discuss the possible selective influence on this tree species, in connection with the identified diversified preferences as to the size (and perhaps condition) of seeds.

 Detailed text notes are included in the accompanying original manuscript file.

Reviewer 3 Report

There are some errors in the work that lead to misinterpretations in the discussion.

78   Natural regeneration is not truncated by rodents because many rodent species are both predators and scatter hoarders. Depending on the annual harvest, they may play one role or the other.

102 There may be an error in the experimental design. As the bare ground plots are placed in the first line from the left, when the mice enter from this position, they find acorns that are easy to find. If sufficient acorns are available, the rodent does not continue searching as it has all the acorns it needs at its disposal. This may encourage the researcher to think that the acorns hidden in the other plots are more difficult to locate when in fact it is because it is already satiated. It would be convenient to alternate the 9 plots in each replicate.

150   It says RIS and it should be PIS

153    It says bare ground and it should be soil burial

154  predation after dispersal PAD is higher in litter cover and soil burial plots than in bare ground, so survival cannot be higher in these plots with higher predation. If predation in situ PIS and predation after dispersal PAD are interpreted equally in figure 3 there is a contradiction.

There is some confusion in the discussion due to figure 3 where there are errors in acronyms.

The percentage of seeds does not clearly show whether it refers to seeds that remain or have been consumed.

In predation in situ PIS indicates that predation on bare ground is greater, therefore the percentage referred to is the number of acorns consumed, however, when referring to predation after dispersal PAD indicates that it is the probability of survival and that this is greater in litter cover and soil burial, therefore here the percentage should refer to the number of acorns remaining in these treatments because if it refers to the number of acorns depredated the figure indicates that more acorns have been eliminated in these treatments than in bare ground.

Of course, seeds forgotten after dispersal can germinate and develop into seedlings. But not only forgotten seeds, the remains of partially consumed acorns can also germinate into seedlings.

There is no reference to partially consumed acorns. Acorns weighing 3 grams are unlikely to be eaten in a single attack, given the size of the rodent. It is likely that the authors have included them as depredated acorns, but some rodents tend to preserve the embryo. In order to assess seed survival, these remains should be included as still active.

The reasoning that increasing the handling time of acorns is risky for the rodent and therefore leads to predation in situ seems to us to be adequate. Digging up acorns increases the handling time and therefore the highest predation in situ occurs on bare ground.

This experiment was conducted without any vegetation cover. The risk of self‐predation is similar in all three treatments, but in nature where there is litter cover, it is possible that in this type of habitat foraging is more intense than on bare ground where there is no protection whatsoever.

On the trade‐off between store o hoard, we think the explanation is correct, but when we talk about evolution, stating that it is more advantageous to hoard than the cost of predation, it should be clearly stated that we are referring to the benefit of the plant, not the rodent.

In periods of high seed production, even if the intensity of predation is high, more seeds can escape than in years of scarcity, when the few seeds produced end up being consumed. For this reason, the reasoning that the lower the density, the more acorns can escape predation does not seem to us to be very accurate. Rodents search more when they cannot find them and predation can wipe them all out, whereas at higher densities they consume more due to predation, but those that are not attacked due to satiation can survive.
